# Feature Cross-Substitution in Adversarial Classification

**Bo Li and Yevgeniy Vorobeychik**
Electrical Engineering and Computer Science
Vanderbilt University
{bo.li.2,yevgeniy.vorobeychik}@vanderbilt.edu

## Abstract

The success of machine learning, particularly in supervised settings, has led to numerous attempts to apply it in adversarial settings such as spam and malware detection. The core challenge in this class of applications is that adversaries are not static data generators, but make a deliberate effort to evade the classifiers deployed to detect them. We investigate both the problem of modeling the objectives of such adversaries, as well as the algorithmic problem of accounting for rational, objective-driven adversaries. In particular, we demonstrate severe shortcomings of feature reduction in adversarial settings using several natural adversarial objective functions, an observation that is particularly pronounced when the adversary is able to substitute across similar features (for example, replace words with synonyms or replace letters in words). We offer a simple heuristic method for making learning more robust to feature cross-substitution attacks. We then present a more general approach based on mixed-integer linear programming with constraint generation, which implicitly trades off overfitting and feature selection in an adversarial setting using a sparse regularizer along with an evasion model. Our approach is the first method for combining an adversarial classification algorithm with a very general class of models of adversarial classifier evasion. We show that our algorithmic approach significantly outperforms state-of-the-art alternatives.

## 1 Introduction

The success of machine learning has led to its widespread use as a workhorse in a wide variety of domains, from text and language recognition to trading agent design. It has also made significant inroads into security applications, such as fraud detection, computer intrusion detection, and web search [1, 2]. The use of machine (classification) learning in security settings has especially piqued the interest of the research community in recent years because traditional learning algorithms are highly susceptible to a number of attacks [3, 4, 5, 6, 7]. The class of attacks that is of interest to us are *evasion* attacks, in which an intelligent adversary attempts to adjust their behavior so as to evade a classifier that is expressly designed to detect it [3, 8, 9].

Machine learning has been an especially important tool for filtering spam and phishing email, which we treat henceforth as our canonical motivating domain. To date, there has been extensive research investigating spam and phish detection strategies using machine learning, most without considering adversarial modification [10, 11, 12]. Failing to consider an adversary, however, exposes spam and phishing detection systems to evasion attacks. Typically, the predicament of adversarial evasion is dealt with by repeatedly re-learning the classifier. This is a weak solution, however, since evasion tends to be rather quick, and re-learning is a costly task, since it requires one to label a large number of instances (in crowdsourced labeling, one also exposes the system to deliberate corruption of the *training* data). Therefore, several efforts have focused on proactive approaches of modeling the

learner and adversary as players in a game in which the learner chooses a classifier or a learning algorithm, and the attacker modifies either the training or test data [13, 14, 15, 16, 8, 17, 18].

Spam and phish detection, like many classification domains, tends to suffer from the curse of dimensionality [11]. Feature reduction is therefore standard practice, either explicitly, by pruning features which lack sufficient discriminating power, implicitly, by using regularization, or both [19]. One of our key novel insights is that in adversarial tasks, feature selection can open the door for the adversary to evade the classification system. This metaphorical door is open particularly widely in cases where *feature cross-substitution* is viable. By feature cross-substitution, we mean that the adversary can accomplish essentially the same end by using one feature in place of another. Consider, for example, a typical spam detection system using a "bag-of-words" feature vector. Words which in training data are highly indicative of spam can easily be substituted for by an adversary using synonyms or through substituting characters within a word (such replacing an "o" with a "0"). We support our insight through extensive experiments, exhibiting potential perils of traditional means for feature selection. While our illustration of feature cross-substitution focuses on spam, we note that the phenomenon is quite general. As another example, many Unix system commands have substitutes. For example, you can scan text using "less", "more", "cat", and you can copy file1 to file2 by "cp file1 file2" or "cat file1 > file2". Thus, if one learns to detect malicious scripts without accounting for such equivalences, the resulting classifier will be easy to evade.

Our first proposed solution to the problem of feature reduction in adversarial classification is *equivalence-based learning*, or constructing features based on feature equivalence classes, rather than the underlying feature space. We show that this heuristic approach does, indeed, significantly improve resilience of classifiers to adversarial evasion. Our second proposed solution is more principled, and takes the form of a general bi-level mixed integer linear program to solve a Stackelberg game model of interactions between a learner and a collection of adversaries whose objectives are inferred from training data. The baseline formulation is quite intractable, and we offer two techniques for making it tractable: first, we cluster adversarial objectives, and second, we use constraint generation to iteratively converge upon a locally optimal solution. The principal merits of our proposed bi-level optimization approach over the state of the art are: a) it is able to capture a very general class of adversary models, including the model proposed by Lowd and Meek [8], as well as our own which enables feature cross-substitution; in contrast, state-of-the-art approaches are specifically tailored to their highly restrictive threat models; and b) it makes an implicit tradeoff between feature selection through the use of sparse ($l_1$) regularization and adversarial evasion (through the adversary model), thereby solving the problem of adversarial feature selection.

In summary, our contributions are:

1. A new adversarial evasion model that explicitly accounts for the ability to cross-substitute features (Section 3),

2. an experimental demonstration of the perils of traditional feature selection (Section 4),

3. a heuristic class-based learning approach (Section 5), and

4. a bi-level optimization framework and solution methods that make a principled tradeoff between feature selection and adversarial evasion (Section 6).

## 2 Problem definition

**The Learner**

Let $X \subseteq \mathcal{R}^n$ be the feature space, with $n$ the number of features. For a feature vector $x \in X$, we let $x_i$ denote the $i$th feature. Suppose that the training set $(x, y)$ is comprised of feature vectors $x \in X$ generated according to some unknown distribution $x \sim \mathcal{D}$, with $y \in \{-1, +1\}$ the corresponding binary labels, where the meaning of $-1$ is that the instance $x$ is benign, while $+1$ indicates a malicious instance. The learner's task is to learn a classifier $g : X \rightarrow \{-1, +1\}$ to label instances as malicious or benign, using a training data set of labeled instances $\{(x_1, y_1), \ldots, (x_m, y_m)\}$.

**The Adversary**

We suppose that every instance $x \sim \mathcal{D}$ corresponds to a fixed label $y \in \{-1, +1\}$, where a label of $+1$ indicates that this instance $x$ was generated by an adversary. In the context of a threat model, therefore, we take this malicious $x$ to be an expression of *revealed preferences* of the adversary: that is, $x$ is an "ideal" instance that the adversary would generate if it were not marked as malicious (e.g., filtered) by the classifier. The core question is then what *alternative* instance, $x' \in X$, will be generated by the adversary. Clearly, $x'$ would need to evade the classifier $g$, i.e., $g(x') = -1$. However, this cannot be a sufficient condition: after all, the adversary is trying to accomplish some goal. This is where the ideal instance, which we denote $x^A$ comes in: we suppose that the ideal instance achieves the goal and consequently the adversary strives to limit deviations from it according to a cost function $c(x', x^A)$. Therefore, the adversary aims to solve the following optimization problem:

$$\min_{x' \in X: g(x') = -1} c(x', x^A). \tag{1}$$

There is, however, an additional caveat: the adversary typically only has query access to $g(x)$, and queries are costly (they correspond to actual batches of emails being sent out, for example). Thus, we assume that the adversary has a fixed query budget, $B_q$. Additionally, we assume that the adversary also has a cost budget, $B_c$ so that if the solution to the optimization problem (1) found after making $B_q$ queries falls above the cost budget, the adversary will use the ideal instance $x^A$ as $x'$, since deviations fail to satisfy the adversary's main goals.

**The Game**

The game between the learner and the adversary proceeds as follows:

1. The learner uses training data to choose a classifier $g(x)$.
2. Each adversary corresponding to malicious feature vectors $x$ uses a query-based algorithm to (approximately) solve the optimization problem (1) subject to the query and cost budget constraints.
3. The learner's "test" error is measured using a new data set in which every malicious $x \in X$ is replaced with a corresponding $x'$ computed by the adversary in step 2.

## 3  Modeling Feature Cross-Substitution

**Distance-Based Cost Functions**

In one of the first adversarial classification models, Lowd and Meek [8] proposed a natural $l_1$ distance-based cost function which penalizes for deviations from the ideal feature vector $x^A$:

$$c(x', x^A) = \sum_i a_i |x_i' - x_i^A|, \tag{2}$$

where $a_i$ is a relative importance of feature $i$ to the adversary. All follow-up work in the adversarial classification domain has used either this cost function or variations [3, 4, 7, 20].

**Feature Cross-Substitution Attacks**

While distance-based cost functions seem natural models of adversarial objective, they miss an important phenomenon of feature cross-substitution. In spam or phishing, this phenomenon is most obvious when an adversary substitutes words for their synonyms or substitutes similar-looking letters in words. As an example, consider Figure 1 (left), where some features can naturally be substituted for others without significantly changing the original content. These words can contain features with the similar meaning or effect (e.g. *money* and *cash*) or differ in only a few letters (e.g *clearance* and *claerance*). The impact is that the adversary can achieve a much lower cost of transforming an ideal instance $x^A$ using similarity-based feature substitutions than simple distance would admit.

To model feature cross-substitution attacks, we introduce for each feature $i$ an equivalence class of features, $F_i$, which includes all admissible substitutions (e.g., $k$-letter word modifications or

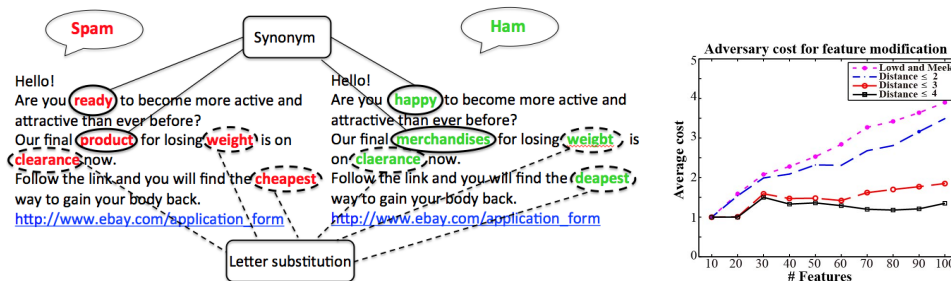

Figure 1: Left: illustration of feature substitution attacks. Right: comparison between distance-based and equivalence-based cost functions.

synonyms), and generalize (2) to account for such cross-feature equivalence:

$$c(x', x^A) = \sum_i \min_{j \in F_i | x_j^A \oplus x_j' = 1} a_i |x_j' - x_i^A|, \qquad (3)$$

where $\oplus$ is the exclusive-or, so that $x_j^A \oplus x_j' = 1$ ensures that we only substitute between different features rather than simply adding features. Figure 1 (right) shows the cost comparison between the Lowd and Meek and equivalence-based cost functions under letter substitution attacks based on Enron email data [21], with the attacker simulated by running a variation of the Lowd and Meek algorithm (see the Supplement for details), given a specified number of features (see Section 4 for the details about how we choose the features). The key observation is that the equivalence-based cost function significantly reduces attack costs compared to the distance-based cost function, with the difference increasing in the size of the equivalence class. The practical import of this observation is that the adversary will far more frequently come under cost budget when he is able to use such substitution attacks. Failure to capture this phenomenon therefore results in a threat model that significantly underestimates the adversary's ability to evade a classifier.

## 4 The Perils of Feature Reduction in Adversarial Classification

Feature reduction is one of the fundamental tasks in machine learning aimed at controlling over-fitting. The insight behind feature reduction in traditional machine learning is that there are two sources of classification error: bias, or the inherent limitation in expressiveness of the hypothesis class, and variance, or inability of a classifier to make accurate generalizations because of over-fitting the training data. We now observe that in adversarial classification, there is a crucial third source of generalization error, introduced by *adversarial evasion*. Our main contribution in this section is to document the tradeoff between feature reduction and the ability of the adversary to evade the classifier and thereby introduce this third kind of generalization error. In addition, we show the important role that feature cross-substitution can play in this phenomenon.

To quantify the perils of feature reduction in adversarial classification, we first train each classifier using a different number of features $n$. In order to draw a uniform comparison across learning algorithms and cost functions, we used an algorithm-independent means to select a subset of features given a fixed feature budget $n$. Specifically, we select the set of features in each case based on a score function $score(i) = |FR_{-1}(i) - FR_{+1}(i)|$, where $FR_C(i)$ represents the frequency that a feature $i$ appears in instances $x$ in class $C \in \{-1, +1\}$. We then sort all the features $i$ according to score and select a subset of $n$ highest ranked features. Finally, we simulate an adversary as running an algorithm which is a generalization of the one proposed by Lowd and Meek [8] to support our proposed equivalence-based cost function (see the Supplement, Section 2, for details).

Our evaluation uses three data sets: Enron email data [21], Ling-spam data [22], and internet advertisement dataset from the UCI repository [23]. The Enron data set was divided into training set of 3172 and a test set of 2000 emails in each of 5 folds of cross-validation, with an equal number of spam and non-spam instances [21]. A total of 3000 features were chosen for the complete feature pool, and we sub-selected between 5 and 1000 of these features for our experiments. The Ling-spam data set was divided into 1158 instances for training and 289 for test in cross-validation with five

times as much non-spam as spam, and contains 1000 features from which between 5 and 500 were sub-selected for the experiments. Finally, the UCI data set was divided into 476 training and 119 test instances in five-fold cross validation, with four times as many advertisement as non-advertisement instances. This data set contains 200 features, of which between 5 and 200 were chosen. For each data set, we compared the effect of adversarial evasion on the performance of four classification algorithms: Naive Bayes, SVM with linear and rbf kernels, and neural network classifiers.

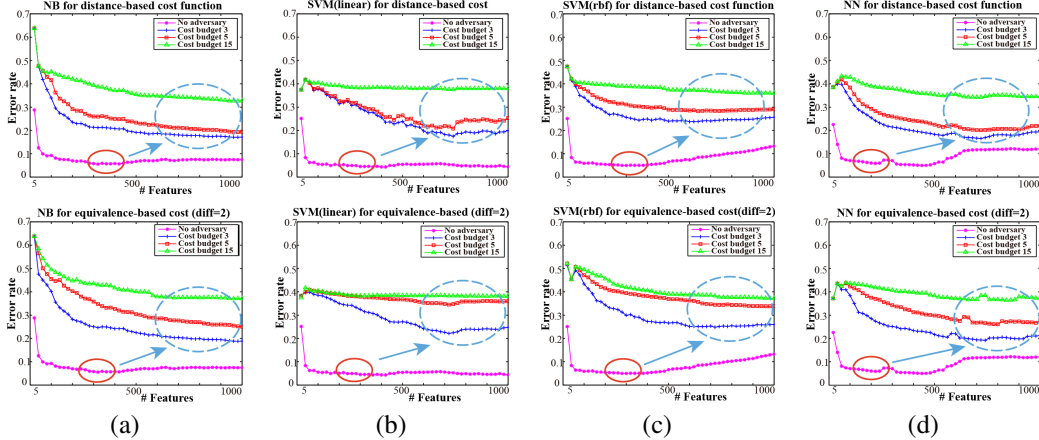

Figure 2: Effect of adversarial evasion on feature reduction strategies. (a)-(d) deterministic Naive Bayes classifier, SVM with linear kernel, SVM with rbf kernel, and Neural network, respectively. Top sets of figures correspond to distance-based and bottom figures are equivalence-based cost functions, where equivalence classes are formed using max-2-letter substitutions.

The results of Enron data are documented in Figure 2; the others are shown in the Supplement. Consider the lowest (purple) lines in all plots, which show cross-validation error as a function of the number of features used, as the baseline comparison. Typically, there is an "optimal" number of features (the small circle), i.e., the point at which the cross-validation error rate first reaches a minimum, and traditional machine learning methods will strive to select the number of features near this point. The first key observation is that whether the adversary uses the distance- or equivalence-based cost functions, there tends to be a shift of this "optimal" point to the right (the large circle): the learner should be using *more* features when facing a threat of adversarial evasion, despite the potential risk of overfitting. The second observation is that when a significant amount of malicious traffic is present, evasion can account for a dominant portion of the test error, shifting the error up significantly. Third, feature cross-substitution attacks can make this error shift more dramatic, particularly as we increase the size of the equivalence class (as documented in the Supplement).

## 5 Equivalence-Based Classification

Having documented the problems associated with feature reduction in adversarial classification, we now offer a simple heuristic solution: equivalence-based classification (EBC). The idea behind EBC is that instead of using underlying features for learning and classification, we use equivalence classes in their place. Specifically, we partition features into equivalence classes. Then, for each equivalence class, we create a corresponding meta-feature to be used in learning. For example, if the underlying features are binary and indicating a presence of a particular word in an email, the equivalence-class meta-feature would be an indicator that *some* member of the class is present in the email. As another example, when features represent frequencies of word occurrences, meta-features could represent aggregate frequencies of features in the corresponding equivalence class.

## 6 Stackelberg Game Multi-Adversary Model

The proposed equivalence-based classification method is a highly heuristic solution to the issue of adversarial feature reduction. We now offer a more principled and general approach to adversarial

classification based on the game model described in Section 2. Formally, we aim to compute a Stackelberg equilibrium of the game in which the learner moves first by choosing a linear classifier $g(x) = w^T x$ and all the attackers simultaneously and independently respond to $g$ by choosing $x'$ according to a query-based algorithm optimizing the cost function $c(x', x^A)$ subject to query and cost budget constraints. Consequently, we term this approach *Stackelberg game multi-adversary model (SMA)*. The optimization problem for the learner then takes the following form:

$$\min_w \alpha \sum_{j|y_j=-1} l(-w^T x_j) + (1-\alpha) \sum_{j|y_j=1} l(w^T F(x_j; w)) + \lambda ||w||_1, \tag{4}$$

where $l(\cdot)$ is the hinge loss function and $\alpha \in [0,1]$ trades off between the importance of false positives and false negatives. Note the addition of $l_1$ regularizer to make an explicit tradeoff between overfitting and resilience to adversarial evasion. Here, $F(x_j; w)$ generically captures the adversarial decision model. In our setting, the adversary uses a query-based algorithm (which is an extension of the algorithm proposed by Lowd and Meek [8]) to approximately minimize cost $c(x', x_j)$ over $x' : w^T x' \leq 0$, subject to budget constraints on cost and the number of queries. In order to solve the optimization problem (4) we now describe how to formulate it as a (very large) mixed-integer linear program (MILP), and then propose several heuristic methods for making it tractable. Since adversaries here correspond to feature vectors $x_j$ which are malicious (and which we interpret as the "ideal" instances $x^A$ of these adversaries), we henceforth refer to a given adversary by the index $j$.

The first step is to observe that the hinge loss function and $||w||_1$ can both be easily linearized using standard methods. We therefore focus on the more challenging task of expressing the adversarial decision in response to a classification choice $w$ as a collection of linear constraints.

To begin, let $\bar{X}$ be the set of all feature vectors that an adversary can compute using a fixed query budget (this is just a conceptual tool; we will not need to know this set in practice, as shown below). The adversary's optimization problem can then be described as computing

$$z_j = \underset{x' \in \bar{X}|w^T x' \leq 0}{\arg\min} c(x', x_j)$$

when the minimum is below the cost budget, and setting $z_j = x_j$ otherwise. Now define an auxiliary matrix $T$ in which each column corresponds to a particular attack feature vector $x'$, which we index using variables $a$; thus $T_{ia}$ corresponds to the value of feature $i$ in attack feature vector with index $a$. Define another auxiliary binary matrix $L$ where $L_{aj} = 1$ iff the strategy $a$ satisfies the budget constraint for the attacker $j$. Next, define a matrix $c$ where $c_{aj}$ is the cost of the strategy $a$ to adversary $j$ (computed using an arbitrary cost function; we can use either the distance- or equivalence-based cost functions, for example). Finally, let $z_{aj}$ be a binary variable that selects exactly one feature vector $a$ for the adversary $j$. First, we must have a constraint that $z_{aj} = 1$ for exactly one strategy $a$: $\sum_a z_{aj} = 1 \forall j$. Now, suppose that the strategy $a$ that is selected is the best available option for the attacker $j$; it may be below the cost budget, in which case this is the strategy used by the adversary, or above budget, in which case $x_j$ is used. We can calculate the resulting value of $w^T F(x_j; w)$ using $e_j = \sum_a z_{aj} w^T (L_{aj} T_a + (1 - L_{aj}) x_j)$. This expression introduces bilinear terms $z_{aj} w^T$, but since $z_{aj}$ are binary these terms can be linearized using McCormick inequalities [24]. To ensure that $z_{ja}$ selects the strategy which minimizes cost among all feasible options, we introduce constraints $\sum_a z_{aj} c_{aj} \leq c_{a'j} + M(1 - r_{a'})$, where $M$ is a large constant and $r_{a'}$ is an indicator variable which is 1 iff $w^T T_{a'} \leq 0$ (that is, if $a'$ is classified as benign); the corresponding term ensures that the constraint is non-trivial only for $a'$ which are classified benign. Finally, we calculate $r_a$ for all $a$ using constraints $(1 - 2r_a) w^T T_a \leq 0$. While this constraint again introduces bilinear terms, these can be linearized as well since $r_a$ are binary. The full MILP formulation is shown in Figure 3 (left).

As is, the resulting MILP is intractable for two reasons: first, the best response must be computed (using a set of constraints above) for each adversary $j$, of which there could be many, and second, we need a set of constraints for each feasible attack action (feature vector) $x \in \bar{X}$ (which we index by $a$). We tackle the first problem by clustering the "ideal" attack vectors $x_j$ into a set of 100 clusters and using the mean of each cluster as $x^A$ for the representative attacker. This dramatically reduces the number of adversaries and, therefore, the size of the problem. To tackle the second problem we use constraint generation to iteratively add strategies $a$ into the above program by executing the Lowd and Meek algorithm in each iteration in response to the classifier $w$ computed in previous iteration. In combination, these techniques allow us to scale the proposed optimization method to realistic problem instances. The full SMA algorithm is shown in Figure 3 (right).

$$\min_{w,z,r} \alpha \sum_{i|y_i=0} D_i + (1-\alpha) \sum_{i|y_i=1} S_i + \lambda \sum_j K_j$$

s.t. : $\forall a, i, j : z_i(a), r(a) \in \{0,1\}$

$$\sum_a z_i(a) = 1$$

$\forall i : e_i = \sum_a m_i(a)(L_{ai}T_a + (1-L_{ai})x_i)$

$\forall a, i, j : -Mz_i(a) \le m_{ij}(a) \le Mz_i(a)$

$\forall a, i, j : w_j - M(1-z_i(a)) \le m_{ij}(a) \le w_j + M(1-z_i(a))$

$\forall a : \sum_j w_j T_{aj} \le 2 \sum_j T_{aj} y_{aj}$

$\forall a, j : -Mr_a \le y_{aj} \le Mr_a$

$\forall a, j : w_j - M(1-r_a) \le y_{aj} \le w_j + M(1-r_a)$

$\forall i : D_i = \max(0, 1 - w^T x_i)$

$\forall i : S_i = \max(0, 1 + e_i)$

$\forall j : K_j = \max(w_j, -w_j)$

---
**Algorithm 1** SMA(X)

---
$T$ =randStrats() // initial set of attacks
$X' \leftarrow$ cluster(X)
$w_0 \leftarrow$ MILP($X', T$)
$w \leftarrow w_0$
**while** $T$ changes **do**
    **for** $x^A \in X'_{spam}$ **do**
        $t$ =computeAttack($x^A, w$)
        $T \leftarrow T \cup t$
    **end for**
    $w \leftarrow$ MILP($X', T$)
**end while**
return $w$

---

Figure 3: Left: MILP to compute solution to (4). Right: SMA iterative algorithm using clustering and constraint generation. The matrices $L$ and $C$ in the MILP can be pre-computed using the matrix of strategies and corresponding indices $T$ in each iteration, as well as the cost budget $B_c$. computeAttack() is the attacker's best response (see the Supplement for details).

## 7 Experiments

In this section we investigate the effectiveness of the two proposed methods: the equivalence-based classification heuristic (EBC) and the Stackelberg game multi-adversary model (SMA) solved using mixed-integer linear programming. As in Section 4, we consider three data sets: the Enron data, Ling-spam data, and UCI data. We draw a comparison to three baselines: 1) "traditional" machine learning algorithms (we report the results for SVM; comparisons to Naive Bayes and Neural Network classifiers are provided in the Supplement, Section 3), 2) Stackelberg prediction game (SPG) algorithm with linear loss [17], and 3) SPG with logistic loss [17]. Both (2) and (3) are state-of-the-art alternative methods developed specifically for adversarial classification problems.

Our first set of results (Figure 4) is a performance comparison of our proposed methods to the three baselines, evaluated using an adversary striving to evade the classifier, subject to query and cost budget constraints. For the Enron data, we can see, remarkably, that the equivalence-based classifier

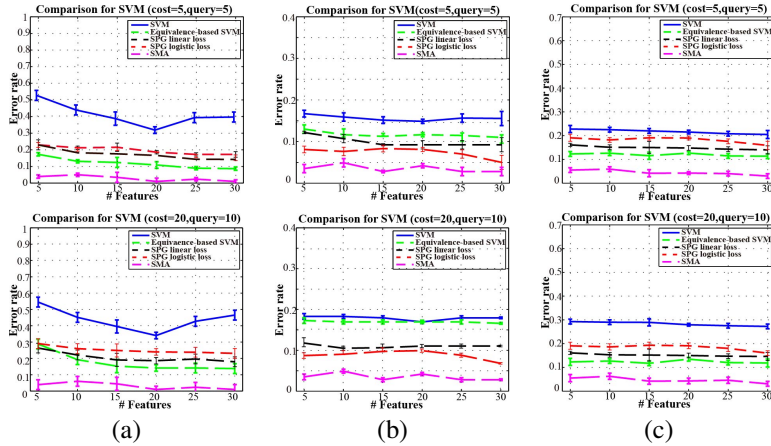

Figure 4: Comparison of EBC and SMA approaches to baseline alternatives on Enron data (a), Ling-spam data (b), and UCI data(c). Top: $B_c = 5, B_q = 5$. Bottom: $B_c = 20, B_q = 10$.

often significantly outperforms both SPG with linear and logistic loss. On the other hand, the performance of EBC is relatively poor on Ling-spam data, although observe that even the traditional SVM classifier has a reasonably low error rate in this case. While the performance of EBC is clearly data-dependent, SMA (purple lines in Figure 4) exhibits dramatic performance improvement compared to alternatives in all instances (see the Supplement, Section 3, for extensive additional experiments, including comparisons to other classifiers, and varying adversary's budget constraints).

Figure 5 (left) looks deeper at the nature of SMA solution vectors $w$. Specifically, we consider how the adversary's strength, as measured by the query budget, affects the sparsity of solutions as measured by $\|w\|_0$. We can see a clear trend: as the adversary's budget increases, solutions become less sparse (only the result for Ling data is shown, but the same trend is observed for other data sets; see the Supplement, Section 3, for details). This is to be expected in the context of our investigation of the impact that adversarial evasion has on feature reduction (Section 4): SMA automatically accounts for the tradeoff between resilience to adversarial evasion and regularization. Finally, Figure 5 (middle, right) considers the impact of the number of clusters used in solving the

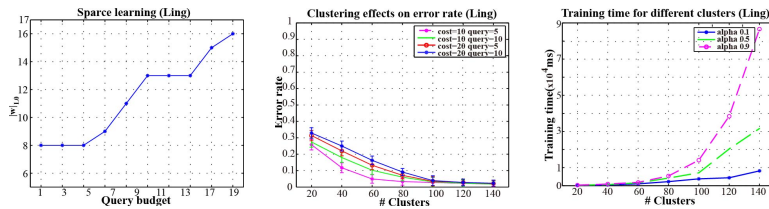

Figure 5: Left: $\|w\|_0$ of the SMA solution for Ling data. Middle: SMA error rates, and Right: SMA running time, as a function of the number of clusters used.

SMA problem on running time and error. The key observation is that with relatively few (80-100) clusters we can achieve near-optimal performance, with significant savings in running time.

# 8 Conclusions

We investigated two phenomena in the context of adversarial classification settings: classifier evasion and feature reduction, exhibiting strong tension between these. The tension is surprising: feature/dimensionality reduction is a hallmark of practical machine learning, and, indeed, is generally viewed as increasing classifier robustness. Our insight, however, is that feature selection will typically provide more room for the intelligent adversary to choose features not used in classification, but providing a near-equivalent alternative to their "ideal" attacks which would otherwise be detected. Terming this idea *feature cross-substitution* (i.e., the ability of the adversary to effectively use different features to achieve the same goal), we offer extensive experimental evidence that aggressive feature reduction does, indeed, weaken classification efficacy in adversarial settings. We offer two solutions to this problem. The first is highly heuristic, using meta-features constructed using feature equivalence classes for classification. The second is a principled and general Stackelberg game multi-adversary model (SMA), solved using mixed-integer linear programming. We use experiments to demonstrate that the first solution often outperforms state-of-the-art adversarial classification methods, while SMA is significantly better than all alternatives in all evaluated cases. We also show that SMA in fact implicitly makes a tradeoff between feature reduction and adversarial evasion, with more features used in the context of stronger adversaries.

### Acknowledgments

This research was partially supported by Sandia National Laboratories. Sandia National Laboratories is a multi-program laboratory managed and operated by Sandia Corporation, a wholly owned subsidiary of Lockheed Martin Corporation, for the U.S. Department of Energy's National Nuclear Security Administration under contract DE-AC04-94AL85000.

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
