[Supplementary Material · Supplement_material.pdf]

# Supplementary Materials for
# *Feature Cross-Substitution in Adversarial Classification*

**Bo Li and Yevgeniy Vorobeychik**
Electrical Engineering and Computer Science
Vanderbilt University
{bo.li.2,yevgeniy.vorobeychik}@vanderbilt.edu

## 1 General feature substitution algorithm for equivalence-based cost function

Here we simulate the behavior of an adversary as running an algorithm $FindBooleanIMAC(x^A, x^-)$ to substitute features from the "ideal" instance $x^A$ based on an arbitrary ham instance $x^-$ to generate the alternative instance $x'$ for the adversary. This is a generalization of the one proposed by Lowd and Meek, which is run only based on the distance-based cost function, to support our proposed equivalence-based cost function.

Within the algorithm 1, function $MatchClass(i, C_v)$ is used to help decide whether it is possible for a feature $i \in C_v$ to be substituted by the others from its class $F_i$, which leads to no cost. Here $C_v$ denotes the vector contains features with different values in $v$ and $x^A$. We employ $MatchClass(i, C_v)$ to guarantee that the number of original substitutable pairs from $x^A$ would not decrease, which leads to a cost of 0. This means we would only change features in $C_y$ that cannot be substituted by features within its class, which is represented as

$$MatchClass(i, C_v) = \sum_{j \in F_i \cap C_v} \mathbb{1}\{f_i \oplus f_j = 1\} - \sum_{j \in F_i \cap C_v} \mathbb{1}\{f_i \oplus f_j = 0\}.$$

Each time the feature is substituted successfully within one iteration, the query count $q$ would increase by 1, until it meets the query budget $B_q$.

**Algorithm 1** $FindBooleanIMAC(x^A, x^-, w, B_q)$

$y \leftarrow x^-$
$flag \leftarrow false$
$q \leftarrow 0$
**repeat**
  $y^{prev} \leftarrow y$
  **for** all $i \in C_y$ **do**
    **if** $F_i \cap C_y = \emptyset$ or $MatchClass(i, C_y) \leq 0$ **then**
      toggle $i$ in $y$
      $q+ = 1$
      **if** $w^T y > TH$ **then**
        toggle $i$ in $y$
        $q- = 1$
      **end if**
    **end if**
  **end for**
  $count \leftarrow 0$
  **for** all $i_1 \notin C_y, i_2, i_3 \in C_y$ **do**
    randomly choose $i_1 \notin C_y, i_2, i_3 \in C_y$ and $i_2 \neq i_3$
    **if** $F_{i_2} \cap C_y = \emptyset$ and $F_{i_3} \cap C_y = \emptyset$; or $MatchClass(i_2, C_y) \leq 0$ and $MatchClass(i_3, C_y) \leq$
    $0$ **then**
      toggle $i_1, i_2, i_3$ in $y$
      $q+ = 1$
      $count \leftarrow count + 1$
      **if** $w^T y > TH$ **then**
        toggle $i_1, i_2, i_3$ in $y$
        $q- = 1$
        $count \leftarrow count - 1$
      **end if**
    **end if**
  **end for**
  **if** $flag$ and $count > 0$ **then**
    $flag \leftarrow false$
  **end if**
  **if** $count = 0$ and $flag = false$ **then**
    $flag \leftarrow true$
    **for** all $i_1 \notin C_y, i_2 \in C_y, i_3 \in C_y$ **do**
      toggle $i_1, i_2, i_3$ in $y$
      $q+ = 1$
      **if** $w^T y > TH$ **then**
        toggle $i_1, i_2, i_3$ in $y$
        $q- = 1$
      **end if**
    **end for**
  **end if**
  **if** $q == B_q$ **then**
    break;
  **end if**
**until** $y^{prev} = y$
return $y$

## 2 Comparison based on different equivalence class sizes

To demonstrate the impact of feature cross-substitution attacks, we show comparisons for NB, SVM with linear kernel, SVM with rbf kernel and Neural Network classifiers based on the baseline Distance-based cost function (Figure 1(a)) and the Equivalence-based cost function (Figure 1 (b)-(d)) cost function with Enron data.

Figure 1: Impacts of different equivalence class sizes for (a) Distance-based cost function, (b) Equivalence-based cost function with max-2-letter substitution, (c) Equivalence-based cost function with max-3-letter substitution, (d) Equivalence-based cost function with max-4-letter substitution.

For the equivalence-based cost function, we applied max-2,3,4-letter substitution respectively to form equivalence classes with increasing sizes. From the comparison results in Figure 1, it is clear that the feature cross-substitution attacks significantly elevate the test error, and such attacks have more impact when the equivalence class size increases.

## 3 Supplementary algorithm for SMA

Within $SMA$ algorithm, attacker strategies are iteratively added into the linear optimization problem through the constraint generation algorithm. Details of algorithm $computeAttack()$ are provided. In essence, this algorithm calls the approximation algorithm for computing a cost-minimizing instance described earlier in the Supplement (findBooleanIMAC()).

---

**Algorithm 2** $computeAttack(x^A, w)$

---

Randomly select $x^-$ from $X$
**return** $FindBooleanIMAC(x^A, x^-, w, B_q)$

---

## 4 Supplymentary feature feduction effects in adversarial classification

Figure 2 provides additional data about the effects of feature reduction in adversarial classification.

Figure 2: Effect of adversarial evasion on feature reduction strategies. (a)-(d) deterministic Naive Bayes classifier, SVM with linear kernel, SVM with rbf kernel, and Neural network, respectively. 1-3 correspond to Enron, Ling-spam, and UCI data sets. Top sets of figures in each case correspond to distance-based and bottom figures are equivalence-based cost functions. For equivalence-based cost functions equivalence classes are formed using max-2-letter substitutions.

# 5 Supplementary Experiments Evaluating SMA Algorithm

In this section, we exhibit the supplementary comparison results to evaluate the effectiveness of the two proposed methods: the equivalence-based classification heuristic (EBC) and the Stackelberg

game multi-adversary model (SMA) solved using mixed-integer linear programming. The evaluation is based on both the distance-based and equivalence-based cost functions.

We employ three datasets: the Enron data, Ling-spam data, and UCI data. Each column in Figure 3 to Figure 8 corresponds to a specific dataset. Figure 3 to Figure 5 show the comparison results for the Stackelberg prediction game (SPG) with linear loss, SPG with logistic loss, the two proposed methods, and each of the baseline classifier: Naive Bayes, SVM, and Neural Network respectively, based on equivalence-based cost function. Figure 6 to Figure 8 reveal similar comparison results for the two SPG state-of-the-art alternatives and the proposed methods with each baseline classifier based on the distance-based cost function. Various cost (5, 10, 20) and query (5, 10) budget constraints are applied to simulate the adversarial evasion.

From Figure 3 to Figure 8, it is evident that SMA outperforms other alternatives in all situations subject to various combinations of cost and query budget constraints based on different datasets. The performance of EBC is relatively data-dependent but still shows resilience to the adversarial feature cross-substitution attacks compared with the traditional baseline classifiers. The comparison results also suggest that given higher cost and query budget, the adversary is better able to perform feature cross-substitution attacks and therefore elevate the test error for the traditional classifiers, which fail to taken adversarial attacks into account. Furthermore, even having considered the adversarial settings for classification tasks, the test error rate of all classifiers based on the distance-based cost function is still higher than the corresponding one based on the equivalence-based cost function. This implies that under estimate the adversary ability would lead to bad performance for classifiers. However, as SMA model can apply more robust cost function (equivalence-based cost function) to evaluate the adversary strategies accordingly during training, the test error of SMA is able to keep relatively stable for different attacked data, which significantly increases the classifier robustness.

Figure 9 takes the insight for the nature of SMA solution vectors $w$ as supplementary comparisons based on both Enron and UCI datasets to the results represented in the main paper, which are based on the Ling-spam data. From the $\|w\|_0$ evaluation based on the two dataset (Figure 9 (a)), we can see the similar trend for different datasets, that as the query budget for adversary increases the solutions become less sparse. Figures 9 (b) and (c) show the clustering effects on test error and training time. Results from different datasets suggest that with more than 100 clusters, the test errors already converges to a near-optimal value, while the training time turns out to be just several seconds. This demonstrates that the SMA model can achieve a fast training process with more accurate classification results compared with alternative classifiers, as well as an automatically trained model to deal with the tradeoff between overfitting and adversarial attacks, including the feature cross-substitution attack.

Figure 3: Comparison of EBC and SMA approaches to the baseline classifier Naive Bayes and SPG alternatives based on Equivalence-based cost function for (a) Enron data, (b)Ling-spam data, and (c) UCI data. Row 1: $B_c = 5, B_q = 5$, Row 2: $B_c = 10, B_q = 5$, Row 3: $B_c = 20, B_q = 5$, Row 4: $B_c = 5, B_q = 10$, Row 5: $B_c = 10, B_q = 10$, Row 6: $B_c = 20, B_q = 10$.

Figure 4: Comparison of EBC and SMA approaches to the baseline classifier SVM and SPG alternatives based on Equivalence-based cost function for (a) Enron data, (b)Ling-spam data, and (c) UCI data. Row 1: $B_c = 5, B_q = 5$, Row 2: $B_c = 10, B_q = 5$, Row 3: $B_c = 20, B_q = 5$, Row 4: $B_c = 5, B_q = 10$, Row 5: $B_c = 10, B_q = 10$, Row 6: $B_c = 20, B_q = 10$.

Figure 5: Comparison of EBC and SMA approaches to the baseline classifier Neural Network and SPG alternatives based on Equivalence-based cost function for (a) Enron data, (b)Ling-spam data, and (c) UCI data. Row 1: $B_c = 5, B_q = 5$, Row 2: $B_c = 10, B_q = 5$, Row 3: $B_c = 20, B_q = 5$, Row 4: $B_c = 5, B_q = 10$, Row 5: $B_c = 10, B_q = 10$, Row 6: $B_c = 20, B_q = 10$.

Figure 6: Comparison of EBC and SMA approaches to the baseline classifier Naive Bayes and SPG alternatives based on Distance-based cost function for (a) Enron data, (b)Ling-spam data, and (c) UCI data. Row 1: $B_c = 5, B_q = 5$, Row 2: $B_c = 10, B_q = 5$, Row 3: $B_c = 20, B_q = 5$, Row 4: $B_c = 5, B_q = 10$, Row 5: $B_c = 10, B_q = 10$, Row 6: $B_c = 20, B_q = 10$.

Figure 7: Comparison of EBC and SMA approaches to the baseline classifier SVM and SPG alternatives based on Distance-based cost function for (a) Enron data, (b)Ling-spam data, and (c) UCI data. Row 1: $B_c = 5, B_q = 5$, Row 2: $B_c = 10, B_q = 5$, Row 3: $B_c = 20, B_q = 5$, Row 4: $B_c = 5, B_q = 10$, Row 5: $B_c = 10, B_q = 10$, Row 6: $B_c = 20, B_q = 10$.

Figure 8: Comparison of EBC and SMA approaches to the baseline classifier Neural Network and SPG alternatives based on Distance-based cost function for (a) Enron data, (b)Ling-spam data, and (c) UCI data. Row 1: $B_c = 5, B_q = 5$, Row 2: $B_c = 10, B_q = 5$, Row 3: $B_c = 20, B_q = 5$, Row 4: $B_c = 5, B_q = 10$, Row 5: $B_c = 10, B_q = 10$, Row 6: $B_c = 20, B_q = 10$.

Figure 9: Left: $\|w\|_0$ of the SMA solution. Middle: SMA error rates, and Right: SMA running time, as a function of the number of clusters used. Top: results based on Enron data. Bottom: results based on UCI data.