[Reviews · NeurIPS 2014]

Submitted by Assigned_Reviewer_4

The paper formalizes the problem of feature cross-substitution in adversarial classification and presents some approaches to solving it (one heuristic and one exact based on mixed-integer linear programming). Previous work in adversarial classification have considered simpler models for how an adversary would try to fool a classifier by replacing values of features (basically, simple distance functions) and assume that the adversary would try to minimize the distance to the real features why fooling the classifier.
The paper points out that this approach has a serious shortcoming when feature selection is done.
The authors then suggest that it makes more sense to consider equivalent classes of features as basically being the same features and treating them transparently in feature selection and classifier building. They show results of their two approaches (heuristic and exact) on three datasets, outperforming previous work.

The paper is overall very well written. I have some suggestions for the figures:
-explain how the graph in figure 1 was generated. Is this for a particular dataset?
-mark the best # of features for each line in fig. 2, to make it easier to compare them as suggested by the text

This paper explains very well previous work and gives clear motivations for the (original) directions they decide to take. Then they show strong experimental results, which I believe are significant for the area. I think they could give more examples of how their approach would work in settings other than spam detection (would the same idea of classes of equivalence of features make sense?)
Summary: This paper is easy to read and follow. It proposes original and technically sound approaches for adversarial classification. I believe this is a significant contribution to this area of research.

Submitted by Assigned_Reviewer_31

The authors propose a cost function to account for adversarial behavior in classification evasion by feature substitution. They then propose two methods to learn a classifier that is robust to adversarial modifications on the test distribution.

The proposed learning algorithm appears to be new and the need for handling general adversarial cost functions is interesting.
While the problem (classification under adversarially changing test distributions) is important, the feature substitution idea may be too domain specific (text data only).
I think the strongest point of the paper is section 6, where the authors explain the main algorithm. However, this section requires significant revision. Specifically, it would be good to have the final formulation of the MILP and the algorithm itself. The usage of defined variables are also not so clear in this form.

Some section specific comments are as follows:

Section 3:
Meaning of the number of features and the way the data is generated is unclear in Figure 1.

Section 4:
The number of plots to make the point is somewhat redundant and the space could have been better utilized in Section 6. It is clear that the probability of one of the removed features being in the equivalence class of one of the used features is increasing as the number of features decrease or the number of words in the equivalence class increase.
Also it seems that generating the data using the equivalence-based cost does not degrade the classifier performance that much for the same cost budget compared to the distance based cost.

Section 6:
A reference to McCormick inequalities would be good.

For the part where you claim the algorithm is intractable in its original form, it would be better to have a complexity analysis.

Did you use the equivalence-based cost as the distance function when clustering the adversarial feature vectors? Based on the experimental results, if you used l2 norm, it would probably mean that there weren’t that many distinct adversaries in your data.

Section 7:
What is the reason that we don’t see the decreasing error rate trend for Ling-spam and UCI data in figure 3 as in figure 2.
Summary: The authors propose a cost function that better models the adversarial behavior in classification evasion by feature substitution. They then propose two methods to learn a classifier that is robust to adversarial modifications on the test distribution.

Submitted by Assigned_Reviewer_36

This paper tackles the problem of classification in an adversarial setting, specifically in the case of spam and phishing detection. It presents a class of adversaries that use "feature cross-substitution" to evade classification while still accomplishing their attacks as well as two approaches to handling this type of adversary: a heuristic one based on feature equivalence classes and a more principled one based on a Stackleberg game solved with integer/linear programming.

Positive Points:

1. The paper is well-written and well-motivated, with a good coverage of the related literature. One possible bit of related work that might be included is that of Karlberger (et al.?) on Penetrating Bayesian Spam Filters.

2. The idea of a complementary form of classification error (c.f. bias and variance) in adversarial learning settings is nice. Probably not an earth shatteringly new/surprising idea, but it is nice to see people thinking about this kind of thing.

3. Empirical results on three different email data sets, compared to state-of-the-art methods, are impressive.

Points of Concern:

4. In Sec. 4, is the simple model of 1- and 2-letter substitutions a reasonable approximation for more sophisticated types of substitution attacks?

5. In Sec. 4, the first key observation, that "when the adversary uses either substitution cost function, the optimal number of features increases", while intuitively appealing, doesn't seem to me to be demonstrated very clearly at all in the reported experiments. The claimed "right shift" does not seem obvious to me in the majority of the results shown in Fig. 2. Indeed, if I were looking at these data in an unbiased way, I'd be tempted to say they show that in fact there is no shift in the majority of cases and, therefore, there is no noticeable tradeoff between overfitting and feature selection under adversarial conditions. Which in turns lead to the question of whether this proposed third type of error really exists. The rest of the paper is pretty clear and convincing (and the empirical results certainly seem to show that something is going on) -- am I missing something here?

6. In Sec. 4, the second key observation seems obvious on the one hand---of course evasion will account for a large amount of error if there is a lot of malicious activity---and, at the same time, I wonder if we can accurately make this claim; that is, are we confident that the bias and variance components of the error really are small under this scenario?

7. Seems like the last paragraph of Sec. 6 is hiding lots and lots and lots of details. I realize NIPS has a tight page limit, but it seems like a bit more here, traded with a bit less somewhere else would make this a better paper.

Small points:

8. References could be tidied up, particularly capitalization issues.

9. In Sec. 2, subheading "The Game", should "the defender" in bullet 3 be reconciled with the "learner" in bullet 1?

10. I'm not clear on what exactly the notation $x^A_j \oplus x^'_j$ means.

11. In Fig. 1, right hand side, does the x-axis label "# Features" refer to the size of the equivalence class (it seems to be hinted that it does in the text, but this should be clarified).

12. There are a few places throughout the paper where there are usage issues with articles (mostly they are occasionally missing).

13. In Sec. 4, when details on the data sets are given, class balance is described for both the Enron and UCI data sets but is neglected for the Ling data set.

14. In Sec. 6, I had a hard time with the definition, $e_j = \Sum…$, where $e_j = w^T…$. Is this just an odd way to say how we are going to compute the $w^TA()$ term in Eq. 4? Or, is something else being said here (with a possible typo obfuscating it)? Can this be clarified somehow?

Interesting questions:

15. With a large enough cost budget, it would seem a foregone conclusion that the defender will have little hope of accurately identifying spam while also allowing legitimate email through. However, it would also seem that at some threshold of substitution, spam effectiveness will begin to really suffer (that is, not even really gullible people will give it any credence at some point). This could be modeled with a non-linear cost function or with a simple threshold, for example, but do we have any way to bound this effective cost budget? That is, is there a point at which we don't have to worry about detecting it because no adversary would choose such a highly substituted attack vector?
Summary: A nice paper with an interesting core idea and nice empirical results to back it up, but with a bit of concern about whether the claimed underlying analysis is correct.
Author Feedback
Author rebuttal: We thank all the reviewers for their thoughtful comments and suggestions.

In particular, we will compress the content for (and eliminate redundancy from) Section 4 and expand Section 6 to improve readability and include the full proposed algorithm. We will also attempt to improve the exposition throughout based on the comments, such as clarifying the axis labels, explaining ambiguous notation (such as x^A_j \oplus x^'_j, as well as e_j definition in Section 6), including relevant information about data sets (class balance for the Ling data set), cleaning up references, etc.

Several reviewers raised the question of generalizability of feature substitution beyond text data.
As another example, many unix system commands have substitutes. For example, you can scan text using less, more, cat. You can copy file1 to file2 by "cp file1 file2" or "cat file1 > file2". And so on. In intrusion detection, if you learn to detect malicious scripts without accounting for such equivalences, the resulting classifier will be easy to evade.

Reviewer 1 (#31):

1. Figure 1 is generated in a way similar to Figure 2, but we show adversarial evasion cost rather than defender's error on the y-axis.

2. The particular impact of going from distance-based to equivalence-based cost function is best seen in the Supplement, Figure 1, going right to left: with a large equivalence set, the adversary can effect a far greater impact for the same budget. In the main section, we restricted attention to 1-/2- letter differences, and the impact is smaller (it tends to be most significant for cost budget = 5, since too small a budget is insufficient for effective attacks in any case, and a large budget admits effective attacks in all cases).

3. The original "naive" representation of the MILP has O(m * 2^n) constraints, where m is the number of positives in data and n the number of features. Thus, even representation is a problem. Running time is worst-case exponential in the size of the MILP, so it cannot be solved in reasonable time for any but toy instances (in theory or in practice). We will clarify this.

4. We used l2 clustering. Our intuition is that, indeed, we do not have too many distinct adversaries in the data, and we are able to successfully exploit this structure.

5. Figures 2 and 3 are not immediately comparable for two reasons: Figure 2 has an infinite query budget, and the x-axis range differs between the two (Figure 3 has a much narrower range). As it happens, in these particular cases over this range, the baseline SVM (rbf kernel) in Figure 3 b&c is relatively flat (see, e.g., Figure 2, 2c, bottom plot, for cost budget = 5, which is somewhat comparable to Figure 3b). In the supplement we provide a number of experiments for alternative values of cost/query budgets, where the rankings of algorithms are quite consistent with what's in the main text, but the shape of the baseline SVM is varying somewhat.

Reviewer 2 (#36):

1. We also considered 3-/4- letter substitutions (see the supplement). The results follow a predictable pattern as the size of the equivalence class increases. We view this as an illustration of the general conceptual point of feature substitution, rather than a full analysis of alternative substitution attacks, which is beyond the scope of our paper, but is certainly an interesting question in its own right.

2. "Figure 2... shows no shift in the majority of cases". We are amiss for not clarifying this. In many cases, there is a relatively flat region for a "naive" classifier. In such a case, we apply a common principle of choosing the smallest # of features that minimizes the cross-validation error. If the figure is viewed through this lens, the distinction becomes more apparent in most cases (at least to us). However, we do not claim universality: there are clearly instances where the error due to adversarial evasion is unimportant for the purposes of feature reduction. For example, this appears to be the case in plot 3(a), top, for cost budget = 5. Our point is merely to identify that this phenomenon exists and often needs to be accounted for.

3. Adversarial error, relative to bias/variance, does depend on the specific setting, and varies significantly in Figure 2. However, we can see that in many cases it is several factors higher (e.g., plots 1(a-d)).

4. Interesting question: there is much to be discussed here. To be brief, we think that substitution model (that is, the definition of equivalence classes) should in practice account for this. Your comment is, indeed, exactly why we introduce the cap on the cost budget as a parameter: there will surely be a cost budget high enough so that an attacker will no longer find it worthwhile to send spam/phish. The question of where such a threshold (cost budget) should be in practice is a good one, and we believe it is still an open question how one can quantify, say, response rate as a function of email content, which would allow one to establish it (recent work on the economics of spam by Stefan Savage's group at UCSD would be helpful here as well).

Reviewer 3 (#4):

We thank you for your suggestions about clarifying Figure 1 and marking the best # of features in Figure 2 to facilitate the comparisons, as well as the suggestion of additional examples for generalizing the approach beyond text-based domains (we offer one such example above).